# Automatic Monitoring Methods for Greenhouse and Hazardous Gases Emitted from Ruminant Production Systems: A Review

**DOI:** 10.3390/s24134423

**Published:** 2024-07-08

**Authors:** Weihong Ma, Xintong Ji, Luyu Ding, Simon X. Yang, Kaijun Guo, Qifeng Li

**Affiliations:** 1College of Animal Science and Technology, Beijing University of Agriculture, Beijing 100096, China; mawh@nercita.org.cn (W.M.); jixintongbua@163.com (X.J.); 2Information Technology Research Center, Beijing Academy of Agriculture and Forestry Sciences, Beijing 100097, China; dingly@nercita.org.cn; 3National Innovation Center of Digital Technology in Animal Husbandry, Beijing 100097, China; 4Advanced Robotics and Intelligent Systems Laboratory, School of Engineering, University of Guelph, Guelph, ON N1G 2W1, Canada; syang@uoguelph.ca

**Keywords:** greenhouse and hazardous gas, ruminant production system, automatic gas monitoring

## Abstract

The research on automatic monitoring methods for greenhouse gases and hazardous gas emissions is currently a focal point in the fields of environmental science and climatology. Until 2023, the amount of greenhouse gases emitted by the livestock sector accounts for about 11–17% of total global emissions, with enteric fermentation in ruminants being the main source of the gases. With the escalating problem of global climate change, accurate and effective monitoring of gas emissions has become a top priority. Presently, the determination of gas emission indices relies on specialized instrumentation such as breathing chambers, greenfeed systems, methane laser detectors, etc., each characterized by distinct principles, applicability, and accuracy levels. This paper first explains the mechanisms and effects of gas production by ruminant production systems, focusing on the monitoring methods, principles, advantages, and disadvantages of monitoring gas concentrations, and a summary of existing methods reveals their shortcomings, such as limited applicability, low accuracy, and high cost. In response to the current challenges in the field of equipment for monitoring greenhouse and hazardous gas emissions from ruminant production systems, this paper outlines future perspectives with the aim of developing more efficient, user-friendly, and cost-effective monitoring instruments.

## 1. Introduction

Livestock contribute about 11–17 percent of total global anthropogenic emissions of greenhouse gases, with ruminants producing 28 percent of total global methane emissions. Ruminants also produce hazardous gases, which have an impact on the environment and animals, contribute to global warming and climate change, adversely affect ecosystems and the climate system, and may pose a significant risk to animal health [1,2,3]. Among them, methane is a potent greenhouse gas, contributing more to global warming than carbon dioxide [4]. Consequently, mitigating measures are imperative to curtail livestock greenhouse and hazardous gas emissions, safeguarding the environment and human health.

Ruminants primarily emit gases into the atmosphere through two main pathways: firstly, greenhouse and hazardous gases such as methane, generated via in vivo fermentation in the gastrointestinal tract, are directly released into the atmosphere through belching, burping, or flatulence [5,6]. Secondly, organic matter such as feces and urine excreted by livestock undergoes in vitro anaerobic fermentation, yielding methane as an indirect emission into the atmosphere. In addition to mitigating greenhouse and hazardous gas production at its source through optimized feeding practices, the monitoring, detection, and capture of gases play pivotal roles. Given that greenhouse and hazardous gases such as methane and carbon dioxide are valuable energy resources, leveraging digital detection methods becomes crucial for assessing the efficacy of emission reduction strategies [7]. In this paper, the monitoring of greenhouse gas concentrations emitted directly from animals is presented separately from the monitoring of hazardous gas emissions in the environment, with a focus on gas monitoring methods.

The present review will primarily focus on the comparative analysis of the monitoring of the concentration and emission of greenhouse and hazardous gas emissions from ruminant animals, examining each method. We will commence by introducing the types of gases emitted by ruminant animals and explaining their production mechanisms. Subsequently, a comprehensive overview and comparison of gas concentration monitoring methods and gas emission monitoring techniques will be provided, exploring their underlying principles and applicability. Finally, the discussion will encompass the challenges faced by greenhouse and hazardous gas monitoring devices and their future developmental trajectories.

## 2. Types and Sources of GHG and Hazardous Gases from Ruminant Production Systems

Ruminant animals are distinguished by their distinctive gastric structure, comprising four primary compartments: the rumen, reticulum, omasum, and abomasum. This gastric configuration enables ruminants to exhibit unique rumination behavior during food digestion. The rumination process involves the initial fermentation of food in the rumen, followed by regurgitation and rechewing in the oral cavity. This process not only aids in a more thorough food breakdown but also allows microbial communities to influence food.

Among these, the concentration of ruminant-generated gases, especially greenhouse gases (GHGs), also plays a key role in our understanding and assessment of the contribution of the agricultural sector to GHG emissions. Nevertheless, these concentrations exhibit variability contingent upon factors such as the species of ruminant (e.g., cattle versus sheep), feeding methodologies (e.g., pasture-based versus concentrate diets), and managerial strategies (e.g., feeding regimens and waste handling practices). The data presented in Table 1 show the greenhouse and hazardous gas concentrations generated per individual ruminant per diurnal cycle. A comprehensive comprehension of the mechanisms producing these greenhouse and hazardous gases significantly augments our capacity to understand the impact of ruminant animal feeding on the environment while facilitating smart management and mitigation of greenhouse and hazardous gas emissions.

### 2.1. Ruminant Greenhouse and Hazardous Gases

Ruminant rumen microorganisms produce large amounts of carbon dioxide, methane, and ammonia, as well as small amounts of hydrogen, hydrogen sulfide, carbon monoxide, and other gases during the fermentation of feed, of which 74.4% of carbon dioxide, 17.3% of methane, and 6.2% of H_2_S are produced as shown in Figure 1 [8,9]. The graph shows us the types of gases emitted by ruminants to the environment and the proportion of that gas in the total gases emitted by animals to the environment, with the remaining components, such as ammonia, hydrogen sulfide, and water vapor, not being able to be presented as specific values because of their low content [10]. On a global scale, methane and carbon dioxide emitted by ruminant animals are substantial contributors to total GHG emissions [5,11].

### 2.2. Mechanisms of Greenhouse and Hazardous Gases Produced by Ruminant Animals

Ruminant animals generate GHGs predominantly through the metabolic activities of microorganisms within their specialized digestive systems, with methane being a primary constituent. This process occurs in the rumen, where microbial fermentation during food digestion produces hydrogen and carbon dioxide as byproducts. Other microorganisms then utilize these gases to generate methane. In addition to methane and carbon dioxide, ruminant animals emit relatively lower quantities of hazardous gases, such as ammonia and hydrogen sulfide. Among the gases produced by ruminants, although NH_3_ and H_2_S are not major greenhouse gases, they are hazardous to the environment and animals and are classified as hazardous gases [13,14]. As greenhouse gas detection equipment can monitor these gases, we will investigate these two gases in the article as well. Other substances produced include nitrous oxide and water, but their environmental impact is minimal due to their small quantities and will not be extensively discussed here.

#### 2.2.1. Carbon Dioxide (Greenhouse Gas)

Carbon dioxide, with the chemical formula CO_2_, is a common gas found in nature and is the primary GHG. It has the capability to absorb infrared radiation emitted by the Earth, leading to an increase in the near-surface atmospheric temperature. Additionally, carbon dioxide acts as an insulator; its buildup in the atmosphere forms an invisible “glass ceiling”, preventing the dissipation of heat from solar radiation absorbed by the Earth’s surface into outer space, consequently raising surface temperatures [15]. Carbon dioxide originates from various sources within ruminant animals: cellulose, the primary source of sugars in ruminant feed, undergoes synergistic action with ciliates and bacteria in the rumen, leading to its further generation and decomposition into volatile fatty acids, methane, and carbon dioxide. Protein feed undergoes hydrolysis by proteolytic enzymes in the rumen to produce free amino acids and peptides, which are subsequently decomposed by microbial deaminases to produce ammonia, carbon dioxide, and volatile fatty acids. A portion of the ammonia produced from amino acid breakdown is absorbed by the rumen epithelium, while the rest participates in the ornithine cycle to produce urea. Urea is then hydrolyzed by urease into carbon dioxide and ammonia, the decomposition process of which is shown in Figure 2 [16].

#### 2.2.2. Methane (Greenhouse Gas)

Methane, with the chemical formula CH_4_, is the simplest hydrocarbon. It is a colorless and odorless gas under standard conditions, widely used as a fuel in daily life and industry [17]. Although carbon dioxide is more abundant in the atmosphere, methane has a higher capacity to absorb infrared radiation than carbon dioxide. Carbon dioxide is one of the more dominant greenhouse gases. However, the structure of the methane molecule is such that it can absorb more infrared radiation at specific wavelengths, particularly in the wavelength range between 3–4 μm and 7.5–8.5 μm, than carbon dioxide. As shown in Figure 3, ruminants convert food into gas by consuming feed in a complex gastric environment (especially the rumen), and the gas is expelled from the mouth and the anus, with anal emission being the main form of excretion, and a small portion being released to the environment in the feces [18]. Starch, protein, and cell walls in the feed are broken down by microorganisms to produce acetic acid, propionic acid, butyric acid, hydrogen, and carbon dioxide. Methanogenic bacteria then convert these gases, particularly hydrogen and formate, into methane through reduction reactions [19,20].

#### 2.2.3. Ammonia (Hazardous Gas)

Ammonia is an inorganic compound with the chemical formula NH_3_ and is a colorless gas with a strong irritating odor. It can cause burns to the skin, eyes, and respiratory organs, and excessive inhalation by humans can cause swelling of the lungs to the point of death [22]. Experiments found that, during the rearing of sheep, the phenomenon of tearing occurs in many pens, and in severe cases, the cornea becomes greyish-white, blocking the line of sight, resulting in blindness and causing the sheep to bump into walls and fences, affecting feeding; lambs have symptoms of sneezing, coughing, wheezing, runny noses, and respiratory mucosal infections in severe cases, causing lung inflammation and leading to coughing, runny noses, and breathing difficulties. After dissection, the lungs have a black-colored area, areas of adhesion to the mucosa of the thoracic cavity, areas of exudation of purulent material, etc. In animal welfare, ammonia concentration levels ranging from 0 to 5 mL are suitable for animal survival. Prolonged exposure to concentrations between 5 and 19 mg/L of ammonia can irritate the eyes and respiratory mucous membranes in animals. Concentrations exceeding 20 mg/L are associated with inflammatory responses, and severe inflammation can lead to mortality [23].

#### 2.2.4. Hydrogen Sulfide (Hazardous Gas)

Hydrogen sulfide, with the chemical formula H_2_S, is an inorganic compound that is flammable and acidic under standard conditions. It is colorless with a rotten egg odor at low concentrations and a sulfurous odor at higher concentrations, and it is highly toxic. When hydrogen sulfide mixes with air, it can form explosive combinations and can ignite and explode when exposed to flames or high heat [24]. Livestock fed sulfur-containing high-protein feed can suffer from digestive disorders, during which a large amount of hydrogen sulfide is discharged from the intestinal tract. This contains sulfide fecal accumulations and decomposition or corruption can produce hydrogen sulfide, in which the concentration of H_2_S must not exceed 25 mg/m^3^ [25]. Hydrogen sulfide can irritate the nasal cavity, leading to rhinitis, and damage the trachea and lungs, resulting in tracheitis and lung edema. It can also lead to vegetative nerve disorders. Hydrogen sulfide in the blood will make the animal’s body hypoxic, weaken livestock, and reduce immunity [26].

## 3. Gas Concentration Monitoring

Greenhouse gas and hazardous gas monitoring methods and concentration techniques encompass electrochemical monitoring instruments founded on redox reactions between gases and electrodes, infrared detection instruments leveraging absorption spectra, such as Fourier transform infrared spectrometers (FTIR), non-dispersive infrared (NDIR) sensors, and laser detectors hinging on the spectral absorption principle between lasers and gas molecules, including handheld laser methane detectors (LMD) and Tunable diode laser absorption spectroscopy (TDLAS). Gas chromatography (GC) operates on the premise that gas phases carrying distinct substances traverse the column at varying velocities. Moreover, more sophisticated monitoring methodologies include remote monitoring via aircraft, drones, and satellites, along with SWIR (short-wave infrared) cameras, as depicted in Figure 4.

Compared to gas emission monitoring methods, gas concentration monitoring equipment has the advantages of smaller monitoring devices, a wider range of application, higher accuracy, and shorter manual operation time. For example, infrared sensors such as TDLAS can identify greenhouse and hazardous gas components in mixed gases and reduce the influence of interfering gases, thereby improving detection accuracy. With the rapid development of sensor technology, such devices’ prices have significantly decreased. FTIR has been successful in continuously monitoring CH_4_ and CO_2_ levels in the atmosphere, but environmental factors such as temperature, atmospheric pressure, and humidity can impact measurement accuracy and precision. Deploying these instruments in more controlled farming environments can mitigate these effects, enhancing their usability.

Recently, aircraft, satellites, and drones have been used in pastures and farms. Satellite and remote sensing technologies for methane and other greenhouse and hazardous gases detection have been applied in industry and agriculture. However, further development is needed in the livestock sector. In conclusion, there is a need for more innovation and dissemination of intelligent detection devices in the ruminant livestock industry.

### 3.1. Chemical Sensors

Chemical sensors are specialized devices designed to detect, identify, and quantify chemical substances. They operate by utilizing chemical reactions, physical changes, or biological recognition to convert the signals from target compounds into measurable electrical, optical, or other signals, thus realizing the detection and analysis of target compounds [27]. They are characterized by high sensitivity, fast responses, selectivity, portability, and high efficiency and reliability, and they are widely used in industrial fields. Their use in the livestock industry is mainly for monitoring the concentration of gases such as NH_3_, CO_2_, and CH_4_ in livestock barns. These gases serve as critical indicators of air quality, and excessive concentrations can adversely affect animal health. The main types of chemical sensors include electronic detectors, gas chromatography, semiconductor metal oxide chemical sensors, and pellistor sensors [28]. Each type has its unique working principle and application scenarios, providing a necessary technical means for chemical analysis and monitoring.

#### 3.1.1. Electrochemical Detectors

Electrochemical gas sensors are powered by batteries and operate based on electrochemical reactions between gases and chemical reagents at the working electrode. These reactions generate electric currents whose strength varies with changes in gas concentration. By detecting the current values and their trends, these sensors can calculate gas concentrations and observe emission patterns and behaviors. There are several types of electrochemical sensors, including primary battery types, concentration cell types, constant potential electrolysis cell types, and limiting current type sensors. These sensors have high sensitivity, selectivity, and stability, allowing them to detect multiple toxic gases simultaneously, such as carbon monoxide, ammonia, hydrogen sulfide, etc., as shown in Figure 5. One promising approach to selectively detecting multiple gases with a single electrode is chemically modifying or functionalizing the electrode surface. This method involves altering the electrode’s chemical composition or introducing specific functional groups or catalysts to enhance its sensitivity and selectivity towards different gases [29,30]. However, this method may have limitations in terms of response time and operational lifespan.

Electrochemical-mediated sensors are mainly used for detecting hazardous gases with electrochemical activity. Compared to catalytic combustion sensors, hazardous gases passing through electrochemical sensors tend to form relatively stable diffusion barriers, which can reduce detection sensitivity. When using such sensors, it is essential to note that some gases lack electrochemical activity, so they may easily undergo cross-reactions with minor components inside the instrument, leading to detection delays and affecting the accuracy of measurement data. Electrochemical sensors are sensitive to environmental factors such as humidity, temperature fluctuations, and corrosive substances such as acids, alkalis, metals, or other gases. Therefore, maintaining a clean detection environment and promptly addressing any deviations in data are essential when using these sensors.

Electrochemical methods involve gas contacting the working electrode to initiate electrode reactions, producing electric signals proportional to the gas concentration. Electrochemical sensors can be divided into potential, conductance, capacitance, and current sensors based on the differences in transmission signals. These sensors obtain gas concentrations by converting induced electromotive force signals, electrolyte solution conductance signals, electrolyte solution and electrode interface capacitance signals, and external circuit current signals. Electrochemical methods are widely applied for detecting NH_3_ and H_2_S in livestock and poultry facilities, and they can also be used for CO_2_ detection.

When measuring NH_3_ concentrations using the electrochemical method, NH_3_ undergoes an oxidation reaction on the working electrode. In contrast, a reduction reaction occurs on the counter (auxiliary) electrode, and a potential signal is formed between the working electrode and the counter electrode that varies with NH_3_ concentration. Real-time NH_3_ concentration data can be obtained by resolving this signal [31]. The chemical reactions occurring on the working electrode and counter electrode of a typical NH_3_ electrochemical sensor are [32].
$$2NH_{3}+6OH^{-}\to N_{2}+6H_{2}O+6e^{-},$$
$$O_{2}+2H_{2}O+4e^{-}\to 4OH^{-}$$

The principle of using the electrochemical method to measure H_2_S concentrations is similar to that of NH_3_, by resolving the redox reaction occurring on the two electrodes in the loop to produce a current signal proportional to the concentration of H_2_S. Then, real-time H_2_S concentration data can be obtained. Regarding metal oxide gas sensors, Royster et al. used tungsten oxide doped with appropriate tungsten carboxylate to produce hydrogen sulfide sensors for detecting low concentrations of H_2_S, with high sensitivity, reusability, and stable performance [33]. Additionally, there are numerous handheld H_2_S sensors on the market that leverage the electrochemical reaction of carbon nanotubes to enhance electrodes made from glassy carbon and carbon fiber. The detection limit of the equipment is 0.3 μmol/L, and it exhibits excellent detection performance within the concentration range of 1.25 μmol/L to 112.5 μmol/L [34]. The chemical reactions occurring on the sensitive and auxiliary electrodes of a typical H_2_S electrochemical sensor are
$$H_{2}S+4H_{2}O\to H_{2}SO_{4}+8H^{+}+8e^{-};$$
$$2H_{2}O\to O_{2}+4H^{+}+4e^{-}.$$

The measurement of CO_2_ concentrations using the electrochemical method can be achieved by using lithium carbonate (Li_2_CO_3_) or barium carbonate (BaCO_3_) as the sensitive electrode and a sodium super-ionic conductor (NASICION) as the solid electrolyte. The chemical reactions that occur on the sensitive electrode and the solid electrolyte are [35,36,37]
$$2Li^{+}(Ba^{2+})+CO_{2}+\frac{1}{2}O_{2}+2e^{-}\to Li_{2}CO_{3}(BaCO_{3}),$$
$$2Na^{+}+\frac{1}{2}O_{2}+2e^{-}\to Na_{2}O.$$

#### 3.1.2. GC

Gas chromatography operates on the principle that different substances in the gas phase move at different speeds through a chromatographic column, facilitating their separation [38]. The column outlet is connected to a detector, where, as components pass through the column sequentially, the sensor records the signal intensity of each element and converts it into an electrical signal. This electrical signal is then amplified, collected in a data collection set, and transmitted to a computer. Subsequently, the electrical signals are further processed in a chromatography workstation to generate a gas chromatogram, where the target gas concentration can be calculated based on the peak height and the chromatographic peak area [39]. Various detectors are used in GC for gas concentration determination, including flame photometric, micro-argon ionization, and thermal conductivity detectors, as depicted in Figure 6 [17]. Early gas chromatographs utilized dosing valves for sample injection, while improved gas chromatographs can inject samples using a syringe, thereby reducing detection costs and improving efficiency [40]. Gas chromatographs have the advantages of high sensitivity, high selectivity, high speed, a wide range of applications, and simple equipment and operation. However, it has environmental limitations, requiring skilled operators, strict environmental conditions, complex sample preparation, and constraints on analytical scope. Additionally, high sample concentrations are often necessary for accurate analysis [38].

#### 3.1.3. Semiconductor Metal Oxide Chemical Sensors

In recent years, sensors based on the principle of semiconducting metal oxides (SMOxs) have gained significant attention due to their sensitivity, stability, and cost-effectiveness as key performance indicators. SMOxs are standard gas sensors used to detect the concentration of gases in the environment. Common metal oxide materials include SnO_2_, ZnO, etc. [41]. The operational principle involves these materials changing their electrical properties when exposed to target gases. Gas molecules come into contact with the sensor and adsorb onto the material’s surface, thus changing the material’s conductive properties [42,43]. These sensors also typically include a heating element to increase the sensor’s sensitivity and allow for the simultaneous detection of many different gases. Semiconductor metal oxide sensors are favored in environmental testing for their rapid response times, affordability, and compact size. The principle of operation of semiconductor metal oxide sensors is based on the daughter-in-law of gas molecules on the surface of the semiconductor metal oxide leading to a change in electrochemical properties, rather than relying on redox reactions of the gas molecules at the electrodes, which is what distinguishes them from electrochemical sensors [44]. However, despite their widespread use in environmental monitoring, these sensors have not yet been extensively applied in the livestock industry for greenhouse gas monitoring. This represents a promising area for further exploration and development.

#### 3.1.4. Pellistor Sensors

Combustion-type sensors, also known as Pellistor sensors, work based on the principle of the chemical reaction of gases with oxygen [45,46]. These sensors detect and measure gas concentrations by monitoring changes in the resistance of a heating element [47,48]. They convert chemical signals generated by the interaction of gases with oxygen into measurable electrical or other signals, enabling the detection and analysis of combustible gases [49].

Using combustion sensors to monitor the presence of combustible gases (e.g., CH_4_, NH_3_, etc.) in barns or livestock feeding environments allows for the timely detection of potential fire or explosion risks and appropriate safety measures to safeguard barns and livestock. It can also monitor the concentration of hazardous gases, including CO_2_ and SO_2_, and changes in oxygen levels. Good air quality helps to improve the production performance and health of livestock. Pellistor sensors have the advantages of fast response time, high sensitivity, and low cost. However, they are less selective for the simultaneous measurement of multiple gases, have a shorter service life, and are susceptible to environmental influences [47]. Despite their benefits and similar to SMOxs sensors, Pellistor sensors have not yet seen widespread adoption in the livestock industry. This presents an area for potential future application and development.

### 3.2. Infrared Sensors

Infrared sensors operate on the principle of spectral absorption, utilizing the unique absorption peaks of gases in the infrared region to detect their presence and concentration. These sensors excel in sensitivity, particularly for hydrocarbons such as methane, ammonia, acetylene, and carbon dioxide [50]. They accurately identify gases based on their absorption characteristics and intensity.

Infrared sensors are a breakthrough from other types of sensors that can be damaged by overheating and can be used in low-oxygen environments, as well as for the hazard detection of corrosive and reactive gases if the gas to be measured is not in direct contact with the sensor. There are also problems such as poor humidity resistance and a complex structure, and the electromagnetic pulse generated by high-power instruments and equipment may also interfere with the sensor [51,52].

#### 3.2.1. FTIR

FTIR identifies and quantifies the composition of substances by measuring the absorption of infrared light at specific wavelengths. The FTIR process begins with infrared light from a source being converted into an interferogram through an interferometer. This interferogram is then directed onto the sample, and the detector captures the resulting infrared spectrum (Figure 7). Computerized Fourier Transform processing converts the interferogram into a spectral map, which reveals the absorption characteristics of the sample [53,54]. The open optical path design can make the path of light in the gas as long as 1 km, and it is widely believed that the FTIR technique is an effective tool for detecting the volume fraction of atmospheric NH_3_, which has been commonly used in atmospheric monitoring [55]. In 1976, Tuazon E C et al. carried out NH_3_ volume fraction measurements in the California region of the U.S.A. They used an optical cell with a length of 22.5 m, consisting of eight mirrors and with an effective absorption path of 2 km. With an effective absorption path of 2 km, the NH_3_ detection limit was 1.3 × 10^−6^. In 2000, Galle B et al. [56] measured NH_3_ volume fractions near ruminant farms in the range of 0.15 × 10^−6^ to 0.25 × 10^−6^ with an effective absorption path of 96 m. In 2023, Lv et al. developed a set of 90 m open optical path Fourier transform infrared spectroscopy (OP-FTIR) measurement equipment for GHG analysis. The experimental results show that the developed OP-FTIR spectroscopy system has high reliability in monitoring the mass concentrations of GHGs. Temperature, relative humidity, wind speed, and wind direction have significant effects on the mass concentrations of local pollutants, and the temporal variations of the mass concentrations of CO_2_, CH_4_, and CO have prominent cyclic trends. Correlations between the mass concentrations of CO and CH_4_, respectively, with that of CO_2_ were analyzed, yielding correlation coefficients of 0.495 and 0.659, respectively [57].

Drones and satellites usually hit on methane and other gas sensors, generally based on the infrared spectrum. Drones and satellites are increasingly employed for quantifying methane emissions stemming from vessels powered by liquefied natural gas, aiming to enhance comprehension of their contributions to the phenomenon of global warming [58]. These technologies leverage high-resolution imagery to assess soil health, monitor crop vitality, and estimate crop yields accurately. Drones are also effective in counting animal populations and detecting methane leaks in natural gas infrastructure. These techniques have been applied on a small scale to assess and determine livestock-related methane emissions on farms, and Vinković et al. [59] have used the innovative UAV-based active AirCore system to accurately measure CH_4_ mole fractions and conduct N_2_O tracer release experiments on a dairy farm in the Netherlands, yielding promising experimental outcomes. However, further research in this area is necessary to broaden the scope and improve methodologies for assessing methane emissions from livestock using aerial technologies.

Advancements in remote sensing and satellite monitoring systems are crucial for accurately quantifying and surveilling methane emissions. Satellite-based measurements offer significant advantages in terms of spatial and temporal coverage, as well as the ability to pinpoint emission hotspots more effectively compared to traditional on-site methods. State-of-the-art sensors now achieve impressive accuracy levels of 0.3 percent, with improved spatial resolution covering 5–10 km and detection precision of within 10 parts per billion [60]. These satellite-based measurements hinge upon inverse modeling techniques to elucidate and quantify methane emissions across regional and global scales [61]. In inverse modeling, satellite-derived atmospheric measurements serve as inputs for deducing the location and magnitude of emission sources and rates (satellite-monitored methane emission images are shown in Figure 8). As scientific and technological means continue to advance, aircraft technologies have made significant progress in monitoring methane; however, there are trade-offs to be made in the areas of coverage and accuracy of remote monitoring [62]. Both aircraft payloads and satellite sensors contribute to the global monitoring of methane concentrations, offering insights into emission patterns and sources. Continuous innovations and improvements in these technologies provide a more effective means of accurately locating and monitoring methane emissions and help to strengthen the regulation and management of greenhouse gas emissions [63].

#### 3.2.2. NDIR

NDIR mainly comprises hardware such as an infrared light source and detector and software algorithms such as signal and data processing. NFIR uses the infrared absorption spectrum produced by CH_4_ in the mid-infrared region at a wavelength of 3.31 μm for concentration detection (Figure 9) [65]. The operational principle of NDIR adheres to the Beer–Lambert law, where the concentration of gases in a sample chamber is determined by measuring the absorption of infrared light at specific wavelengths. Here is how it works: an infrared light source emits a beam through a chamber containing the gas sample. The gas molecules absorb infrared light at characteristic frequencies, and the detector measures the amount of absorbed light, correlating it with the concentration of the gas component [66,67]. The infrared light source is generally selected as low-frequency electrically modulated, high-frequency modulated, and steady-state light sources, and the application of coherent light sources such as quantum cascade lasers (QCLs) is now being investigated. Pyroelectric-type detectors and other thermal effects based on the infrared thermal detector are the most widely used. However, they are now gradually applied to the photoelectric impacts based on the infrared photon detector, with a higher detection rate. Algorithms must be proposed for the interference of other gases and the influence of temperature and humidity to reduce the measurement error.

The introduction of neural network support vector machines, multi-parameter model least squares, and other algorithms has helped to correct the nonlinear error caused by temperature, humidity, pressure, gas overlap absorption, random noise, and other factors. Using algorithms such as neural network support vector machines and multi-parameter model least squares, nonlinear errors induced by temperature, humidity, pressure, gas overlap absorption, and random noise were corrected. The gas concentration measurement results before and after algorithmic compensation significantly enhance sensor accuracy and stability. The NDIR sensor developed by Zhang et al. can detect CO_2_ gas with a maximum measurement error of less than ±0.15% at a temperature of 0~30 °C and a detection accuracy of 3%, and has the advantages of high accuracy, an extensive volume range, miniaturization, and good stability [68].

#### 3.2.3. TDLAS

Tunable diode laser absorption spectroscopy (TDLAS) employs a laser with stable intensity and high coherence that can be tuned to specific wavelengths by adjusting its temperature and current through a drive system. This technology utilizes the infrared absorption spectrum of methane (CH_4_) in the near-infrared region, particularly at a wavelength around 1.66 μm. For high-concentration measurements (≥1%), direct absorption detection technology is used. For trace gas detection, such as methane at lower concentrations, wavelength modulation spectroscopy (WMS) combined with harmonic detection techniques is employed [69,70]. Currently, the primary focus of research lies in employing harmonic technology to detect faint signals containing concentration information, thereby reducing detection limits. This is achieved through a combination of hardware and software techniques, including precise control and compensation of laser temperature and current, gas chamber design using hollow-core photonic crystal fibers, digital lock-in amplifier algorithm design, the design of least squares temperatures and humidity compensation algorithms, and harmonic signal processing algorithms. These methodologies aim to extract useful trace concentration information while suppressing system noise interference and mitigating the effects of temperature and humidity, ultimately enabling high-precision, high-sensitivity methane detection [52]. In the TDLAS instrument experiment used by Zhang et al. for measurement, the linear correlation coefficient R-square of the instrument reached up to 0.999, and the fluctuation of the CH_4_ concentration was less than 0.17 ppm in 3.5 h. Liu et al. used a distributed feedback laser with a central wavelength of 1653.72 nm as the detection light source, realized the real-time methane concentration through the micro-control unit’s main control chip, and proposed fuzzy control based on fuzzy control. They proposed a fusion algorithm of wavelength-modulated and direct absorption spectra based on the fuzzy control S-type affiliation function for the inversion of methane concentration data. The relative error range of the sensor for the full-scale test is −5.28~3.67%, which indicates that it has high accuracy for full-scale measurements [40].

#### 3.2.4. Portable InGaAs Laser Methane Detector (LMD)

LMD [71] uses in situ sampling of animal respiration, but it can be used to calculate total ruminant gas emissions (g/d). LMD uses infrared absorption spectroscopy, and similar detection systems are used in other industries [72]. The indium gallium arsenide (InGaAs) second harmonic detection signal is consistent with the two more vital infrared absorption bands (3.3 and 7.6 μm) of CH_4_, and the absorption spectrum of CH_4_ gas approximates the Lorentzian linear equation, which in turn calculates the CH_4_ concentration [73]. The instrument is usually placed at a distance of one to several meters to ensure that it does not interfere with the behavior of the cattle and also to avoid contamination by other animals in the vicinity. The instrument focuses on the density of CH_4_-emitting smoke plumes and is expressed as a result of CH_4_ concentration. Laser spectrometry allows information on CH_4_ concentration to be extracted independently from physiological activities such as ruminating, feeding, and sleeping in cows. Other factors affecting the accuracy of the instrument are primarily related to its application to grazing animals, such as wind speed and direction, relative air humidity, and atmospheric pressure, all of which may have a significant effect on the final CH_4_ concentration [74]. As an example, wind speed was negatively correlated with CH_4_ concentration (r = −0.41). Methane concentrations in the animal’s oral cavity vary over a wide range, depending on the animal’s respiration, inhalation, and air movement, with higher CH_4_ concentrations likely to occur during inhalation relative to lower concentrations during exhalation. The main disadvantage of this technique is that, similar to the sniffer, only the concentration and not the flux is measured. In a confined chamber, measuring the difference in gas concentrations between the interior and exterior, coupled with assumptions regarding the geometric parameters of the chamber (such as volume and surface area), temperature, and the application of the ideal gas law, facilitates the estimation of gas flux. This method serves as an effective means of assessing the transport rate and diffusion characteristics of gases under specific environmental conditions, thereby establishing a quantitative foundation for the analysis of gas flow processes. However, there are significant limitations to using the LMD in outdoor breeding animals, as air movement can dilute the concentration of the gas being measured to the point where detection is compromised. In one study, CH_4_ measurements by LMD were in good agreement with those in the respiratory chamber (r = 0.8). Still, they did not give good results in most experiments; therefore, despite the ease of use of hand-held lasers on commercial farms, there is still a need to improve their accuracy for practical use [75].

SWIR, known in full as short-wave (length) infrared (band), generally has a wavelength range of 0.85–2.5 μm, and the main sensing material is InGaAs (Indium Gallium Arsenide) [76]. SWIR cameras are typically ground-mounted as imaging spectrometers. They share operational principles with the Laser Methane Detectors (LMDs) used alongside aircraft and satellites to identify previously unknown leaks and quantify emission rates with detailed plume images. Most SWIR detectors are manufactured using complex semiconductor processes, leading to reduced sensitivity at room temperature due to inherent thermal noise issues [77].

The SWIR camera is mounted on a rotating ground-based platform to scan the target area horizontally. It captures skylight from a shallow elevation angle and records the SWIR spectrum between short-wave infrared (SWIR, 950 nm to 2500 nm), while the wind measurement lidar also measures wind direction and speed at different heights. The spatial resolution is about 0.8 m × 0.8 m and a single SWIR scan takes about 60 s. Scans are gathered continuously at 1-min intervals over a 4-day period. The resulting images are augmented with wind, temperature, pressure, and other meteorological data. Recent measurements have shown a minimum sensitivity of 10 ppm for methane near 3.3 μm in less than 10 s (Figure 10) [78,79].

### 3.3. Laser Spectral Detectors

This technique uses the properties of laser light and the spectral absorption phenomenon that occurs between methane gas molecules to measure the concentration of methane gas. As the laser beam passes through the gas to be measured, the methane molecules absorb laser energy at a specific wavelength. This absorption phenomenon is the only thing that can distinguish methane from other gas molecules. Specifically, a laser methane detector contains a laser source, an optical path system, a sensor, and a signal processing module inside [80]. The laser source generates a laser beam of a specific wavelength, which is collimated by the optical path system and then irradiated into the gas to be measured. If methane molecules are present in the gas, the methane molecules will absorb a portion of the laser energy, and the detector will receive the remaining laser beam. The detector converts the received optical signal into an electrical signal, then amplifies, filters, and digitizes it using the signal processing module to finally obtain the methane gas concentration value. Therefore, the sample’s methane gas concentration can be determined by measuring the change in light intensity after the light beam passes through the sample [81,82].

The portable poultry house air quality high-sensitivity monitoring equipment was designed and developed independently by the Beijing Academy of Agriculture and Forestry Sciences to solve the problems of the poor detection limit and sensitivity of the current equipment and the more severe cross-sensitivity. The development of this device uses a laser gas sensor; it can achieve the multi-parameter detection of methane, carbon dioxide, ammonia, temperature, humidity, TVOC, etc. It detects substances at very low levels (sub-ppm) with a minimal error of ±0.1 ppm. The development of this device is based on research into the principles governing detector response drift and its compensation methods, the study of surface emissivity variation effects and mitigation strategies, and the theoretical foundations of miniaturized, low-cost, high-precision infrared thermal imaging sensors. It can monitor the concentration of ammonia and other concentrations in the chicken house in a lower concentration range of trace jitter. The device includes self-developed laser gas sensors (NH_3_/CO_2_), dust sensors (PM_10_/PM_2.5_/PM_1.0_), temperature and humidity sensors, wind speed sensors, odor sensors, and the uploading of environmental parameters on the main board. It can be used for carbon dioxide, methane emission source inspection, and positioning. The performance of this device far surpasses that of electrochemical and semiconductor sensors and compares favorably with mainstream foreign products, with some metrics even being superior.

Methane sensors carried on board aircraft are usually designed based on the principles of laser absorption spectroscopy. In European countries such as the UK and France, aircraft and satellites assisted in a research campaign to measure methane emissions as permafrost thaws north of the Arctic Circle [83]. In South Korea, scientists have used aircraft and drones to measure urban emissions of GHGs that have already liquefied natural gas leaks, with promising results. Such technologies can also be employed in large-scale farming operations [84]. For example, when monitoring methane emissions on a dairy farm, a series of concentric closed flight paths are used, and emission rates are estimated using the Gaussian theorem. As the airplane flew, a series of concentric closed flight paths around the farm facility, methane mixing ratios, pressures, temperatures, and horizontal winds were measured at the barn level to calculate methane emissions from the entire facility. Aircraft measurements were compared to open measurements from an inverse diffusion model, and vehicle measurements were taken using a tracer flux ratio methodology at a California dairy to assess methane emission rates across the farm and at farm-significant emission sources such as animal feeding and liquid manure lagoons. These monitoring techniques are sensitive to capturing methane emission dynamics under different management systems, such as liquid versus dry manure storage systems, which directly affect GHG inventories and climate action. The results of Sahar et al.’s study, which demonstrated that aircraft-based methane sensing technology has matured on the basis of previous research and is ready to play an increasingly important role in environmental policy and regulation, also suggest that the use of aircraft in GHG monitoring will enter a new phase in the context of animal husbandry [85].

## 4. Gas Emissions Monitoring

The main methods for monitoring gas emissions include respiratory chambers, respiratory masks, and Greenfeed. The basic principle of respiratory chambers is that gas samples emitted by ruminants are collected in a single device and analyzed quantitatively and qualitatively using gas analysis techniques to enable monitoring and assessment of gas emissions. Breathing chambers can monitor gases emitted from the animal’s mouth and anus, whereas breathing masks can only monitor gases emitted from the mouth. Although the principle of the breathing mask is similar to that of the breathing chamber, the breathing mask is not as comprehensive as the breathing chamber in terms of gas monitoring. The Greenfeed fully automated gas emission monitoring system is a specialized system for the determination of gas emissions. The scientifically validated Greenfeed system has been used in top international journals and conference abstracts around the world, and researchers have been able to identify and analyze the gas emissions in a number of different ways [86].

### 4.1. Breathing Chambers

Respiration chambers are a well-established and highly accurate method for measuring methane emissions. These chambers are typically categorized as airtight, semi-open, or open systems. Compared to other measurement methods, respiration chambers provide the most precise results because they capture all methane emissions from animals (Figure 11) [87]. Gas analysis instruments are used to analyze the gas samples collected from these chambers. Common methods for gas analysis include infrared spectroscopy, gas chromatography, and chemical sensors. These techniques enable researchers to measure and analyze the components present in the gas samples, thereby determining the gas emissions from ruminants accurately. However, this method has its limitations. Changes in metabolic rates, such as gluconeogenesis, ketosis, or lipid genesis, may be constrained when animals are placed in respiration chambers [88]. Another limitation of respiration chambers is that animals may not exhibit normal behavior; for example, feed intake may decrease, leading to an underestimation of actual methane emissions compared to freely grazing animals under farm conditions [89]. Open-circuit respiration chambers of varying complexity and sizes are widely used, resulting in differing costs. Developing the multi-chamber automated “Respiratory Heat Measurement Series Device for Livestock and Poultry” in China has filled a gap in this field, providing a modern experimental research platform for animal energy metabolism and carbon and nitrogen emission reduction [90,91]. Respiration chambers utilize indirect calorimetry, focusing on the exchange of gases such as oxygen, carbon dioxide, and methane. They can be either open-circuit chambers, which analyze the composition of incoming and outgoing air, or closed-circuit chambers, which analyze the composition of air accumulated over a specific period. In some studies, transparent polycarbonate is used to construct respiration chambers, which include the use of panels with polymethylmethacrylate (acrylic) windows or metabolic crates covered with transparent polycarbonate walls. These respiration chambers are intended to provide accurate and precise measurements of CH_4_ concentrations, including monitoring of animal hindgut emissions. However, these methods are not widely available due to high cost and technical requirements. Not only that, but the gases that can be monitored in the respiratory chamber include CO_2_, NH_3_, CO, nitrogen oxides, and water vapor [86].

One drawback of respiration chambers is that animals may exhibit altered behavior, such as decreased feed intake, which can lead to an underestimate of actual methane CH_4_ emissions compared to animals freely grazing on farms. In research settings, animals are typically subjected to metabolic or performance trials, where trained animals are transferred into the chambers, and CH_4_ measurements are conducted continuously for 3 to 5 days [92]. There are several important factors to consider when conducting controlled experiments, including, but not limited to, gas recovery, routine maintenance, laboratory temperature maintained at less than 27 °C, relative humidity maintained at less than 90%, carbon dioxide concentration maintained at less than 0.5%, ventilation rates in the range of 250 to 260 L/min, and a constant gas flow as recommended by Pinares-Patiño and Waghorn [93].

### 4.2. Breathing Masks

The breathing mask is a device worn over the mouth and nose of ruminants to detect gases, while the head box method involves placing the animal’s head inside a detection box (Figure 12). The latter method, superior to the breathing mask method, allows uninterrupted detection during eating and drinking. However, both methods share common drawbacks: they cannot detect gas discharged from the intestinal tract, and the amount of detection is limited.

A British company, Zelp, has developed a “muzzle” for cattle, utilizing a wrap-around band principle to ensure the animal’s comfort while collecting gas. This innovative mask captures methane from cattle during breathing and burping, reducing emissions by 60 percent. The operational principle involves a fan powered by a rechargeable solar battery that absorbs methane from the cow’s burps. Once the filter in the muzzle becomes saturated, a catalyst converts the methane into carbon dioxide and water. While carbon dioxide is also a GHG, it is less hazardous to global warming than methane. This highlights a compromise and drawback of this equipment: while it effectively reduces methane emissions, it does so by converting methane into another greenhouse gas that still contributes to climate change, albeit to a lesser extent.

### 4.3. Head Breathing Chamber with Hopper

The Greenfeed emissions monitoring system is a self-contained head chamber with a hopper (or, in some cases, two hoppers) above the head chamber. Once the animal’s head is in close proximity to the sensor (utilizing an RFID ear tag identification system), the system is programmed to deliver a small amount of “bait” feed [95]. This feed entices the animal to position its head closer to the sensor, allowing the system to measure increased concentrations of CH_4_ and CO_2_ compared to ambient air levels, as well as gas flow and airflow rates due to the animal’s respiration. To obtain accurate measurements, the head must be positioned for at least 3 min, and when the animal leaves, the system automatically stops gas circulation to end the experiment. A reasonable estimate of methane production from the Greenfeed emission monitoring system depends on the animals accessing the system in a distributed mode over a 24-h cycle, typically with a single experiment lasting 4–7 days [96,97].

The Greenfeed instrument works based on infrared monitoring principles such as NDIR and FTIR. The Greenfeed fully automated gas emission monitoring system is specialized for determining intestinal gas emissions (Figure 13). The scientifically validated Greenfeed system has been used in top international journals and conference abstracts around the world, and researchers have made significant scientific advances and breakthroughs and have been awarded numerous patents [6,86]. This system is applicable for measuring metabolic gases from all cattle and other ruminant animals. It accurately measures carbon dioxide flux levels exceeding 500 g/d and methane flux levels of approximately 3 g/d or more. Under normal operating conditions, 30–40 animals can be measured per day per system, with each animal requiring a certain number of days of measurements. Each animal requires a certain number of days of continuous monitoring [98]. In operation, the system also offers the selective installation of oxygen sensors, hydrogen sensors, dual-feed hoppers, outdoor wind speed and direction sensors, stainless steel outdoor armor, low-temperature adaptation modules, external monitoring cameras, and various other service modules. These options cater to the diverse needs of different ranch conditions [99].

## 5. Discussion

Based on the detailed introduction of various greenhouse and hazardous gas detection methods and instruments mentioned above, Table 2 briefly summarizes the principles and classifications and the advantages and disadvantages of these methods. This table can help readers quickly understand the operational processes, applicable environments, and economic benefits of various instruments, thus enabling them to make more rational choices for the instruments that best suit their needs.

The recent literature underscores significant opportunities for improving greenhouse and hazardous gas detection devices, focusing on their usage, principles, and practical applicability [100]. Modernized livestock farms widely employ industrial sensors; however, their detection accuracy needs to meet the requirements for monitoring air concentrations within poultry and livestock housing. Advanced detection instruments entail high costs, rendering them unsuitable for small-scale farms or individual livestock producers and impeding environmental conservation efforts. Existing greenhouse and hazardous gas monitoring devices deployed in livestock facilities exhibit varying degrees of accuracy and stability issues, affecting the precision of monitoring outcomes. Moreover, most monitoring equipment necessitates operation and maintenance by skilled technicians, yet the need for more qualified professionals constrains the advancement of greenhouse and hazardous gas monitoring in livestock operations.

Cost is a crucial factor when selecting farming products. It directly affects how profitable and sustainable farming operations are. Choosing lower-cost products can lead to competitive pricing and reduce resource waste and environmental impact. Currently, equipment for monitoring greenhouses and hazardous gases is not widely used in farming. The industry still needs to prioritize environmental protection. Therefore, it is important to opt for cost-effective products to promote monitoring methods. For small and medium-sized farmers, LMDs and Laser spectral detectors are recommended. These methods are both high-performance and affordable compared to other options.

## 6. Summary and Future Work

### 6.1. Summary

In animal husbandry, the use of greenhouse and hazardous gas detection equipment holds promise for significant technological advancements, automation, integration, multifunctionality, green environmental protection, and sustainable development. However, realizing these advancements faces challenges in technology, economics, and market dynamics. Gas monitoring devices play a crucial role in monitoring greenhouse and hazardous gases, especially methane emissions from animals. Key challenges include the need for improved accuracy, particularly in methane detection, which is critical for precise monitoring. Stability is also a concern as methane detection tools often experience drift and instability during continuous use, affecting the reliability of monitoring results. Another challenge is the portability of existing devices, which are often bulky and difficult to move, limiting their use across different locations within animal facilities. Furthermore, there is a clear need for greater automation in methane and carbon dioxide detection tools, as many currently rely on manual operation, resulting in slower monitoring processes and reduced accuracy. As environmental concerns grow and sustainable development becomes more critical, there is an opportunity to enhance environmental monitoring within livestock and poultry facilities. This can be achieved by adopting more intelligent and automated monitoring technologies for methane, carbon dioxide, and other greenhouse gases. The goal is to create healthier and safer environmental conditions that support the optimal growth and well-being of animals.

### 6.2. Future Work

Therefore, manufacturers of methane detection equipment in animal husbandry and relevant research institutions need to closely monitor market demands and technological trends. They should enhance cooperation and innovation to drive continuous development and refinement of methane detection technology in animal husbandry. Current equipment used for detecting greenhouse and hazardous gases in animal farming has room for improvement. In the future, detection equipment should prioritize innovation, portability, and cost-effectiveness to better serve the needs of the livestock industry. As international carbon cycle strategies become more sophisticated, there will be increased scrutiny of the carbon footprint of the livestock industry. This will drive the demand for greenhouse and hazardous gas detection equipment in livestock facilities.

## Figures and Tables

**Figure 1 sensors-24-04423-f001:**
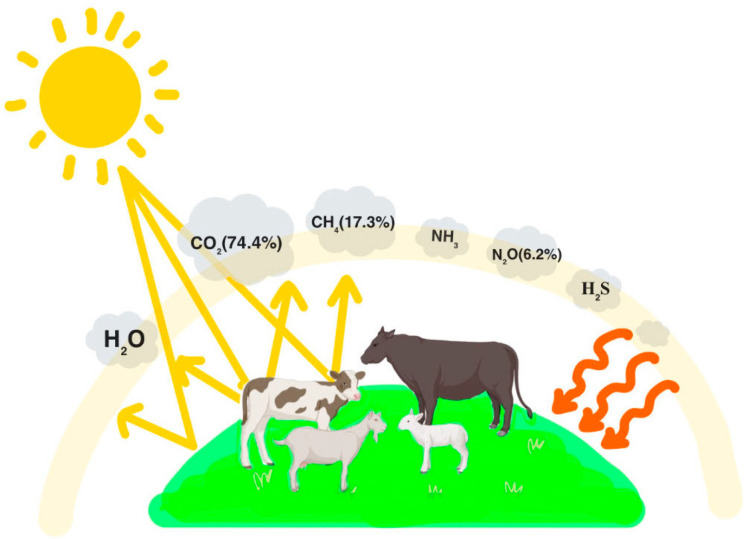
Constituents of gases emitted to the atmosphere by ruminants and their proportions [12].

**Figure 2 sensors-24-04423-f002:**
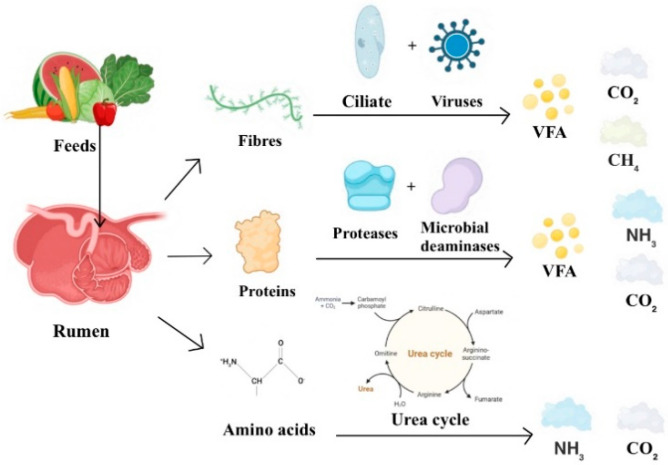
Processes of gas production (mainly CO_2,_ CH_4_, and NH_3_) from feed decomposition in ruminants (mainly in the rumen).

**Figure 3 sensors-24-04423-f003:**
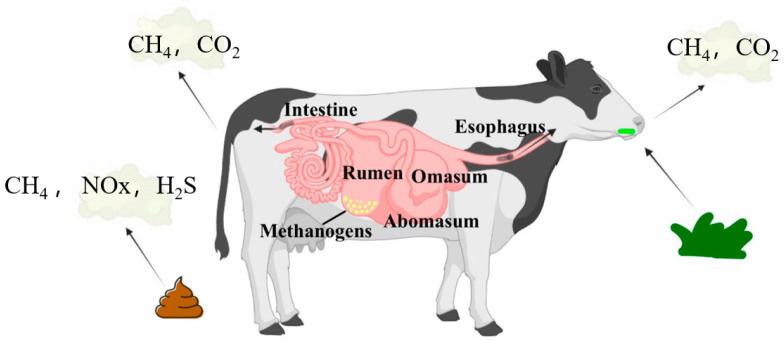
Pathways by which ruminants convert feed into gas for expulsion from the body [21].

**Figure 4 sensors-24-04423-f004:**
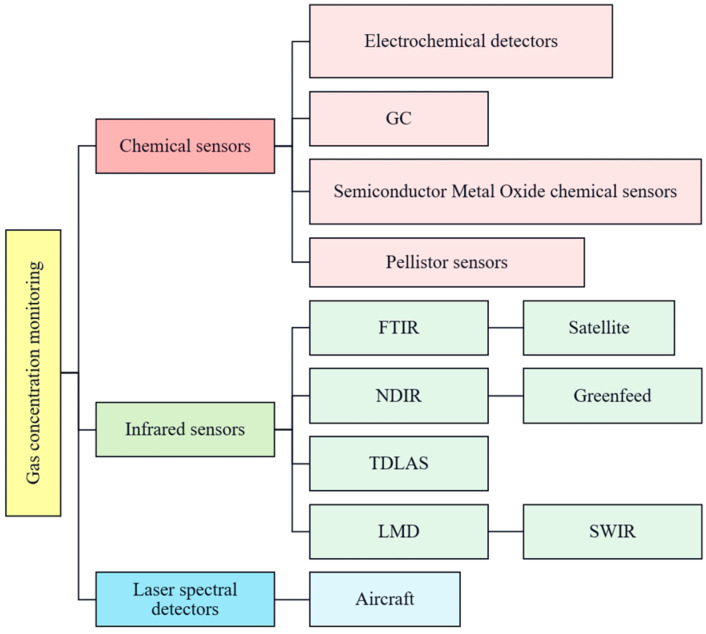
Summary of methodologies for monitoring greenhouse and hazardous gases.

**Figure 5 sensors-24-04423-f005:**
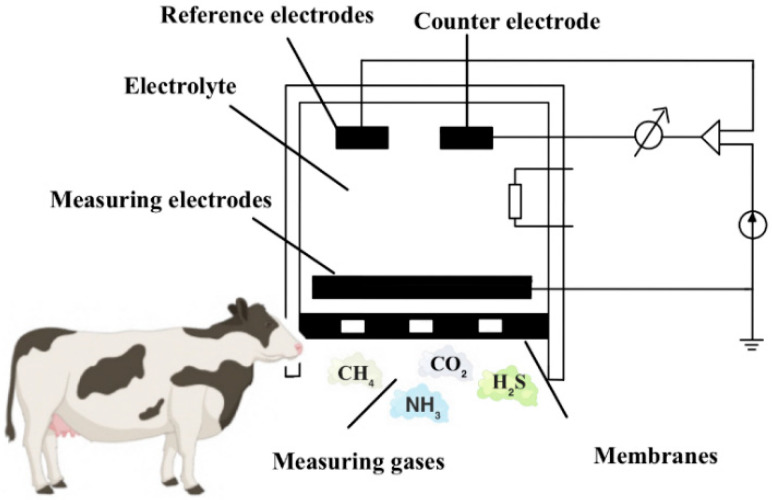
Electrochemical detector detection principle.

**Figure 6 sensors-24-04423-f006:**
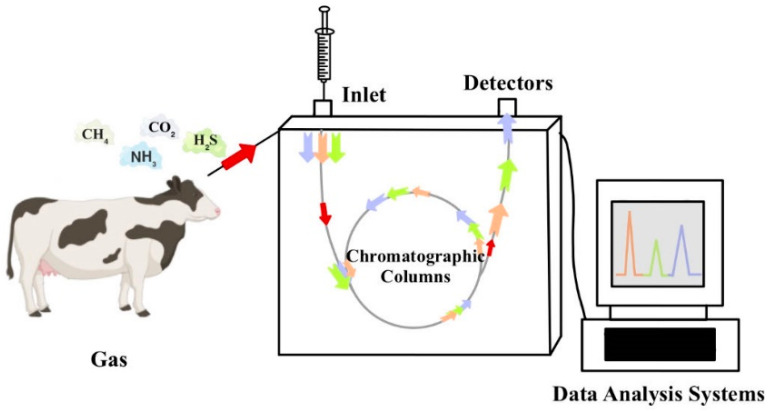
Detection Principle of GC (Gas Chromatography).

**Figure 7 sensors-24-04423-f007:**
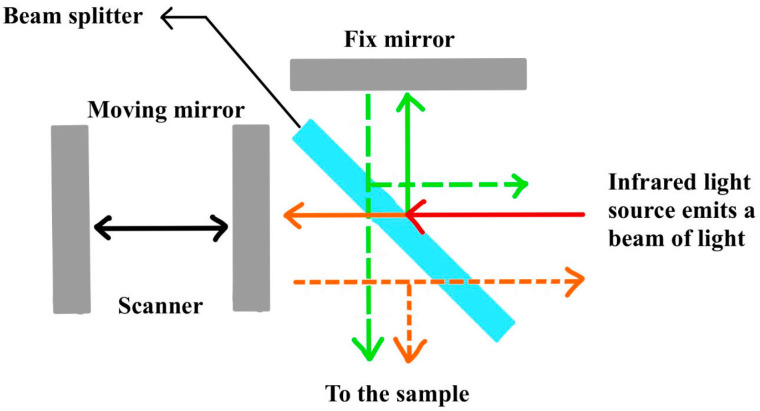
Principles of greenhouse gas monitoring using FTIR detectors.

**Figure 8 sensors-24-04423-f008:**
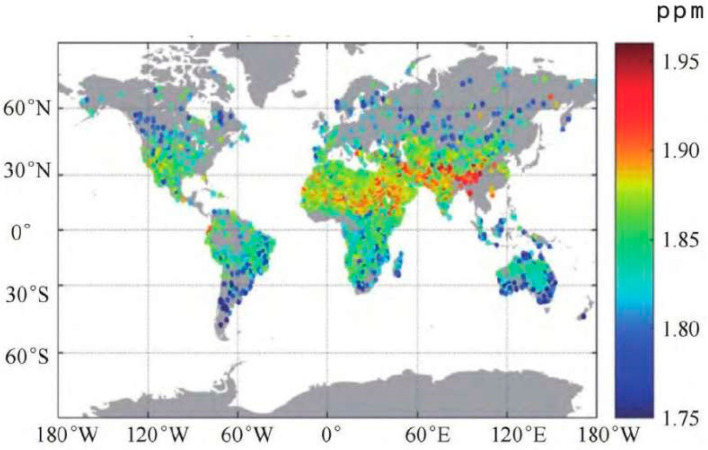
Schematic representation of the effect of satellite monitoring of CH_4_ images [64].

**Figure 9 sensors-24-04423-f009:**
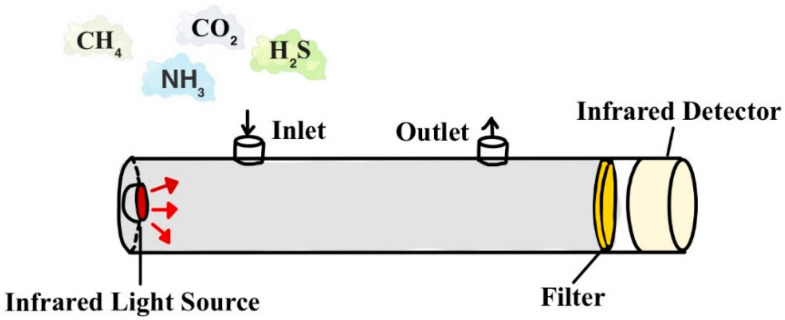
Principles of greenhouse gas monitoring using NDIR detectors.

**Figure 10 sensors-24-04423-f010:**
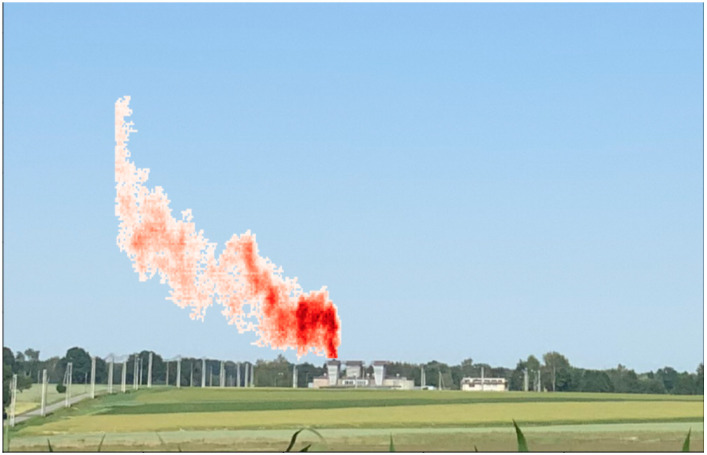
Methane emission plume observed by a SWIR camera, where darker colors indicate higher methane concentrations.

**Figure 11 sensors-24-04423-f011:**
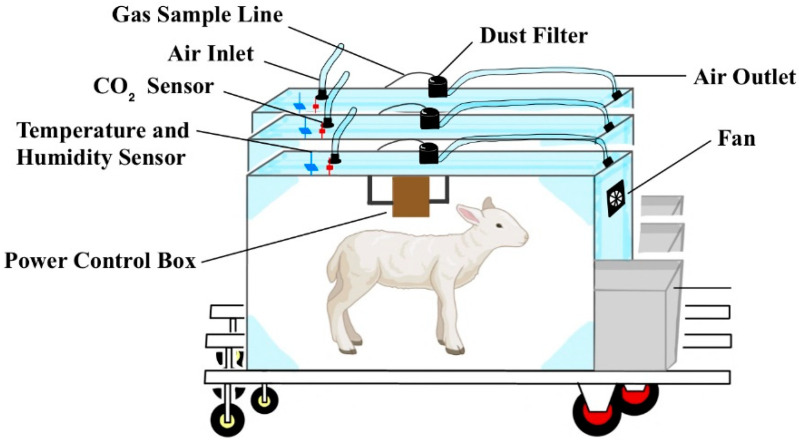
Schematic diagram of the structure of the respiratory chamber and the life of the sheep in the respiratory chamber.

**Figure 12 sensors-24-04423-f012:**
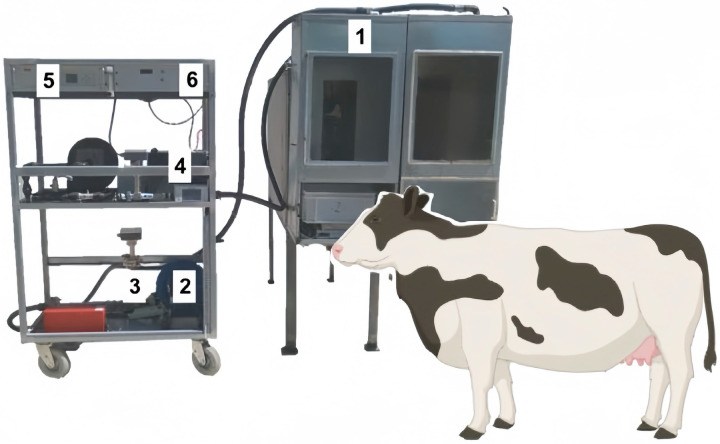
Mobile open-circuit indirect calorimetry equipment cart. (1) Head hood, (2) fan, (3) mass flowmeter, (4) gas cooler, (5) gas analyzer (oxygen, carbon dioxide, and methane), and (6) box for system control and data acquisition panel [94].

**Figure 13 sensors-24-04423-f013:**
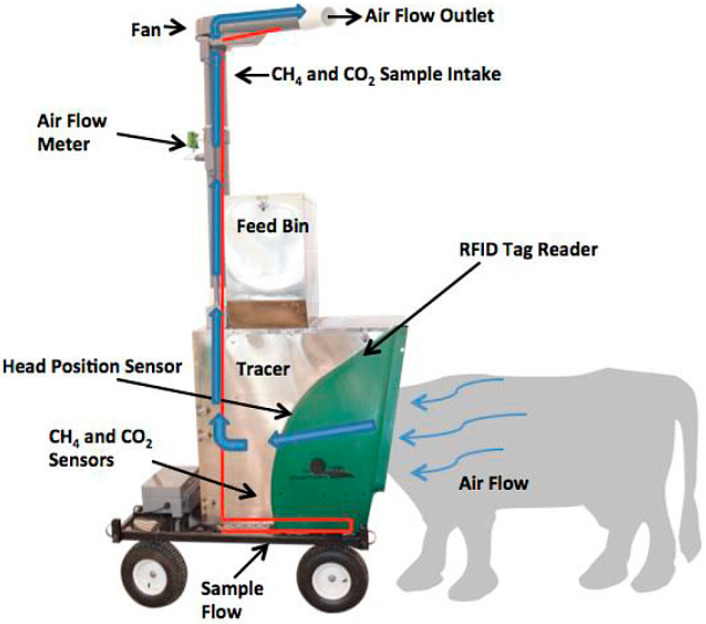
Greenfeed machine model diagram [86].

**Table 1 sensors-24-04423-t001:** Average concentration of greenhouse and hazardous gases emitted by ruminants.

Compounds	Average Concentration (g/d/Animal)
CH_4_	70–120
CO_2_	700–900
NO_x_	5–10
NH_3_	200–400
CO	20–40

**Table 2 sensors-24-04423-t002:** Advantages and disadvantages of methods for monitoring concentrations and emissions of greenhouse gases and harmful gases.

Categories	Principle	Types	Advantages	Disadvantages
Gas concentration monitoring	Measurement of the concentration of gas expelled from the animal.	Electrochemical detectors	Multi-gas non-specific detection High sensitivity	Long response time
			High precision [29]	Short service life [37]
		GC	High efficiency Reliable	Professionally operated
			Multi-sample collection [38]	Higher operating cost
				Gas leakage during use
		SMOxs sensors	High sensitivity	Short service life
			small volume	Sensitivity to environmental factors [44]
			Lower operating costs	
		Pellistor sensors	High sensitivity	Short service life
			Lower operating costs	Sensitivity to environmental factors
				Higher selectivity [47]
		FTIR	Multi-gas non-specific detection [50]	Higher operating costs
		NDIR	Multi-sample collection [50]	Inability to monitor enteric emissions
			Long-term continuous monitoring	Circadian rhythms need to be considered
				time-consuming [68]
		TDLAS	High sensitivity	Higher operating costs [99]
			High precision	
			High anti-interference capability [70]	
		LMD	High sensitivity	Professionally operated
			No individual differences	Sensitivity to environmental factors
			Remote real-time monitoring	Need to keep a distance from animals [72]
		Laser spectral detectors	High sensitivity	Professionally operated
			High precision	
			Lower operating costs	Not tested in the mass market
Gas emission monitoring	Measurement of the amount of gas emitted from environmental and animal fecal deposits	Breathing chambers	Most accurate results [87]	Professionally operated
				time-consuming [86]
			Collection of all gases	Sensitivity to environmental factors
				Influence on animal behavior
				Less efficient experiments
				Higher operating costs [92]
		Breathing mask	Easy to use	Higher error
				Influence on the daily behavior of animals [94]
			Low cost	Inability to monitor enteric emissions
		Head breathing chamber with hopper	Automatic animal recognition [86]	Higher operating costs [92]
			High sensitivity	

## Data Availability

No new data were created or analyzed in this study. Data sharing is not applicable to this article.

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
