# Peer review of "Automatic Monitoring Methods for Greenhouse and Hazardous Gases Emitted from Ruminant Production Systems: A Review"

_sensors, 2024, doi:10.3390/s24134423_

Round 1

Reviewer 1 Report

Comments and Suggestions for Authors

This manuscript provides an overview of measurement techniques to evaluate GHG emissions, particularly methane, from the livestock industry. Some parts are well written and informative, others are highly repetitive and use language apparently generated by an online translator or generative AI. There is very limited new information in the manuscript, and a relatively short reference list that does not justify naming the contribution a "review".

I suggest to reject the manuscript in its current form due to these concerns and invite the authors to resubmit an actual review and analysis if they are so inclined.

Specific comments

1. While I appreciate the overview of in-vivo chemistry and of measurement techniques, very little of the manuscript is intellectually new at all. In essence, Figure 1 and Table 1 represent the basic idea, but they are not discussed in detail in lieu of lots of (distracting) fluff in the rest of the manuscript.

If re-submission occurs, the authors should (i) focus on Fig. 1 and Table 1, (ii) provide adequate and exhaustive citations of previous works to justify naming their contribution a "review", and (iii) establish an evidence-based rapport for the critical columns of "advantages" and "disadvantages" in table 1. That would be a useful contribution to the scientific literature.

2. Mistakes / wrong statements

(i) NH3 and H2S are not greenhouse gases in the common sense

(ii) Figure 2 is misleading as it is unclear whether the numbers in the gas trace clouds are supposed to represent ruminant emission composition or the atmospheric role of the trace gas in the total/natural or man-made greenhouse effect. The reference is not helping.

Figure 4 is useless in its current form.

(iii) the numbers "26" and "22" in lines 146/147 are neither accurate, nor well explained, nor is the provided reference relevant

(iv) the chemical formulae in line 311-312 are not balanced, aka wrong. the chemical equations in line 330 show oxygen with a sup-set "2" (exponent) instead of a sub-set 2; the first also describes CO2 as a product while the sensor is supposedly reacting with CO2 (educt), which is not explained

(v) permafrost thaws, not "melts"

3.  Repetitiveness

In various parts of the manuscript, words and whole text sections are highly repetitive, suggesting lack of proofreading at best.

Examples include page 6, paragraphs before Fig. 5, and section 4.2.2 before Fig. 9. Figures 8, 9, and 10 show essentially the same thing but do not explain the principle adequately.

In-text explanations are at times how one would explain the science to a lay person (e.g. why a gas absorbs IR light), but ought to be scientific, or assume facts to be known to the reader of a scientific journal. They (i.e. IR absorption) are repeated in various ways and thus represent superfluous text.

4. Citations

Several cited works are not easily accessible, but more importantly, the manuscript does not include an exhaustive literature review as would be expected of a "review", and to assist others in selecting a technique for monitoring or spot-checking,

In some cases, such as on page 15, line 564f., an airborne study is described but no citation is given. Similar in lines 355f. In other cases, the provided citation is simply wrong, not related to the statement in the text, e.g. "[46]" in line 448 or "[51]" in line 474.

Comments on the Quality of English Language

The text contains sections and wording that require editing for language, e.g. lines 160, 177, 179, 319-321, 358-359, 360-361 and 392 (wrong names),  519, and in Table 1.

Reviewer 2 Report

Comments and Suggestions for Authors

In the review entitled “Intelligent monitoring methods of greenhouse gas emissions from ruminants: a review”, W. Ma et al. have reported mechanisms and implications of GHG production by ruminant animals, emphasizing a comparative analysis of monitoring methodologies in terms of principles, applicability, and technological aspects.

Please, define which is the percentages of GHG emitted/produced by ruminant animals in the global context, emphasizing the impact, adding references.

Please, briefly, define which is the chemical composition in percentages of GHG produced by ruminant animals, adding references.

Please, improve the quality of Figure 1.

Please improve the manuscript layout: e.g. write the chemical reactions separately from the text, and each for line.

Please, correct the chemical reaction at page 9: “O2” instead of “O2”.

In my opinion, the manuscript can be accepted with minor revision.

Reviewer 3 Report

Comments and Suggestions for Authors

The review is OK.

The section on sensors lacks reference to Semiconductor Metal Oxide chemical sensors and combustion type (pellistor) sensors.

The issue of cost is very important. The lower cost yet high-performance sensors should be emphasized in the last section.

Please improve the English

Comments on the Quality of English Language

Many sentences are phrased in such a way that the meaning is unclear.

The English should be edited.

Reviewer 4 Report

Comments and Suggestions for Authors

“Intelligent monitoring methods of greenhouse gas emissions from ruminants: a review” is a review, that has an classical design for this type of article and it is included introduction; main part with Types of ruminant GHG monitoring and generation mechanisms, Closed and Open monitoring methods; Discussion and conclusion. This review discussed main method of GHG measurement applying to ruminant, while the article provides some interesting information in this field, some details could be improved.

1. Please, add this state “Until 2023, emissions 16 from livestock accounted for over 30% of the total global emissions, with enteric fermentation in 17 ruminant animals being the predominant source of greenhouse gases (GHGs).” into introduction part and add a reference.

2. line 39-47, What is average concentration of the GHGs from ruminant?

3. Figure 2. Not all gases have a content (%). Please, add content (%) for H2S and NH3.

4. Figure 3. In the name of fig formula gases should be given with subscript.

5. For section “2.2.3. Ammonia” and “2.2.4. Hydrogen Sulfide” please, add concentration of H2S and NH3, which lead to described symptoms.

6. Section “3.1. Breathing chambers”. I think a calorimetry principle of GHG detection should be discussed in more detail. Is this device detected only CH4? The same comment for section “3.2. Breathing masks”.

7. lines 311 and 312 need to be formatted.

8. How does device (fig. 7) separate gases (NH3 and H2S) and provide selective measurement? In the both cases are used carbon electrodes?

9. line 312 there is no balance for H in the reaction. The same problem for reaction on the line 311.

10. line 330, I suppose the right parts for reaction is O2 (oxygen generation)? Please, check the reaction.

11. I think into discussion part it should be added table with commercial devices with concentration ranges of GHGs and other analytical parameters.

The manuscript are provide interesting information and could be published after minor revision.

Reviewer 5 Report

Comments and Suggestions for Authors

1.The annotation format of pictures is not uniform, as shown in Figure 8 and Figure 9.
2.Chapter 4 Open monitoring methods is inconsistent with 4.5 Remote monitoring technology format.
3.The formula font is too large and does not match the text font.
4.Figure 6 is not centered.
5.The format of the references is inconsistent, with some missing paper volume numbers.

1. Passive Wireless Detection for Ammonia Based on 2.4 GHz Square Carbon Nanotube-loaded Chipless RFID-inspired Tag. DOI: 10.1109/TIM.2023.3300433

Comments on the Quality of English Language

Moderate editing of English language required

Round 2

Reviewer 1 Report

Comments and Suggestions for Authors

The manuscript has not improved much since the last submission. I maintain my major concern that this is a paper-mill manuscript. You ought to look carefully at the author names and activities, who your reviewers are, and what role the special edition editor plays; and read up on how to guard against such "work", e.g. here:

https://www.wiley.com/en-us/network/publishing/research-publishing/editors/how-can-editors-detect-manuscripts-and-publications-from-paper-mills

I read through, and based on my findings, I maintain my judgement that this manuscript should be rejected.

Any serious content editor, in this case the special edition editor, should not let this even get to the peer review stage in this format.

Comments on the Quality of English Language

Extensive editing of English language required

Author Response

Thank you for your message and the feedback provided on my manuscript.

1. We have taken the reviewer's comment about repetitive content seriously and have already made significant revisions to address this issue. The manuscript now reflects substantial modifications compared to its previous version.

2. To further enhance the quality of the manuscript, We sought guidance and feedback from  professor at the University of Guelph.

3. While we have made efforts to ensure clarity and accuracy throughout, there were instances where I encountered challenges with specific sections, prompting me to utilize online translation tools for assistance.

4. We want to clarify that at no point did I employ AI-generated content in the manuscript. All content was written and revised manually to uphold academic integrity and originality.

Reviewer 5 Report

Comments and Suggestions for Authors

After modification, this manuscript is now available for acceptance

Author Response

Thank you very much.